# Instrument Development and Validation to Assess Care Barriers for Patients in Saudi Arabia with Oral Clefts

**DOI:** 10.3390/ijerph18073399

**Published:** 2021-03-25

**Authors:** Layla Khogeer, Narmin Helal, Osama Basri, Sara Madani, Abeer Basri, Azza A. El-Houseiny

**Affiliations:** 1Department of Pediatric Dentistry, King Abdulaziz University, Jeddah 21589, Saudi Arabia; Nhilal@kau.edu.sa (N.H.); aalhosseiny@kau.edu.sa (A.A.E.-H.); 2Dentistry Department, King Faisal Hospital and Research Center, Jeddah 21499, Saudi Arabia; obasri@gmail.com; 3Dentistry Department, Vision College, Jeddah 23643, Saudi Arabia; sarah-almadani@hotmail.com; 4Dentistry Department, National Guard Hospital, Jeddah 21423, Saudi Arabia; basriab@ngha.med.sa; 5Department of Pediatric Dentistry, Alexandria University, Alexandria 21526, Egypt

**Keywords:** validation, cleft lip and palate, barriers

## Abstract

The study objective was to construct and validate a tool to assess, measure, and evaluate the barriers and obstacles that patients with orofacial clefts (OFCs), and their families, face during treatment. The Effective Accessibility and Accommodation subscale, based on the translated Primary Care Assessment Survey and Primary Care Assessment Tool scales, was used as a reference for the questionnaire. A total of 165 parents from three main cleft referral centers in Saudi Arabia were interviewed. Questionnaire content validity was conducted by calculation of a content validity index for each item (I-CVI) as well as for the total scale (S-CVI). Reliability was tested using Cronbach’s alpha. Factor analysis and principal components analysis were performed to determine the factor structure of the instrument. The final questionnaire had nine items. Rating results showed both I-CVI and S-CVI scores of 1 and Cronbach’s alpha was 0.86. There were three factors (geographic accessibility, appointment availability and accessibility, and scheduling-related barriers) with eigenvalues above 1.00, which collectively accounted for 73% of the variance. In conclusion, this tool is valid and reliable to evaluate accessibility and barriers to care of patients with OFCs in Saudi Arabia.

## 1. Introduction

Orofacial clefts (OFCs) are major human birth defects that represent a significant public health burden [1]. When we look into the literature, we find that there is a scarcity of epidemiological data on cleft lip and palate and other birth defects in many countries around the world. This is despite many efforts made to document the frequency of such defects [2]. In Saudi Arabia, their incidence, based on a review of different studies done in the kingdom, ranges from 0.3 to 2.19 per 1000 live births. OFCs are considered the most common congenital facial malformation [3]. OFCs contribute substantially to long-term social difficulties, complications related to inadequate nutrition, feeding problems, and speech impairments [4,5]. The treatment is a long-term process starting soon after birth, and may continue well up to the end of the second decade of life—with multiple surgeries and long-term medical and dental care [6].

It is of the utmost importance to provide specialized medical care to these children in order to improve their wellbeing and quality of life. In order to achieve ideal outcomes, patients must receive care from multiple different specialists such as pediatric dentists, oral surgeons, otolaryngologists, plastic surgeons, and speech therapists. The treatment received must be in an organized and routine manner [7]. Tertiary care centers are distributed all across Saudi Arabia. However, poor access to care can result in delayed surgical repair of OFCs, leading to poor functional outcomes. It is of vital importance that the healthcare providers assess and evaluate care availability, accessibility, and barriers that may affect patients during the treatment process as these factors may affect the outcome of the overall treatment and prognosis. In order to understand and evaluate the availability and accommodation of perceived oral healthcare needs for cleft lip and palate patients in Saudi Arabia, a valid and comprehensive instrument is one of the main requirements. 

The Effective Accessibility and Accommodation (EAA) subscale is a practical, valid, and reliable measure for patients to evaluate the accessibility of first-contact health services, yielding valid comparisons between urban and rural contexts [8]. No precedent has been found for the use of a tool in Saudi Arabia that measures and evaluates the barriers and obstacles that patients with oral clefts, along with their families, face throughout the treatment process. Although the EAA subscale is a reliable tool, it cannot be used for Arabic-speaking patients with cleft lip and palate conditions. The tool is not designed for this population, hence we had to develop a new tool [8].

The construction and validation of research tools has been frequently done when professionals have observed the need to measure and discuss difficulties and barriers, and no tools are available to faithfully measure these events. The lack of a tool in the literature to assess the obstacles that patients and their families face, not only encouraged this study, but also made it original. Hence, the objective of this study was to present a tool that was constructed and validated to assess, measure, and evaluate the barriers and obstacles that patients with oral clefts, along with their families, face throughout the treatment process.

## 2. Materials and Methods 

### 2.1. Subjects

The sample for this study was recruited through two approaches. First, all the cleft cases with ongoing treatment regardless of their treatment phase visiting the three main Saudi Arabian cleft referral centers from January to March 2019 were included. The main centers were: King Faisal Specialized Hospital and Referral Center (KFSHRC) in Riyadh, which has more than 1000 beds and covers the treatment of most of Saudi Arabia’s cleft cases [1]; KFSHRC in Jeddah; and King Abdulaziz University Hospital in Jeddah, which is an educational and tertiary center for cleft cases. Second, to improve generalizability, a convenient sample was recruited from tertiary centers distributed around Riyadh (King Saud Medical City, Riyadh National Guard Hospital, and King Fahad Medical City) and Jeddah (King Fahad Hospital King Abdulaziz Medical City, King Fahad Armed Hospital, and the Madinah Maternity and Children’s Hospital, which is the only referral cleft center in Madinah). These centers cover most of the cleft cases in Saudi Arabia and are well distributed geographically (Table 1). Ethical approval was obtained from the Research Ethics Committee, Faculty of Dentistry, King Abdulaziz University (166-12-18). The subjects were contacted by phoning the primary caregiver number listed in the patients’ records. Parents consented verbally to participate in the study and their participation was completely voluntary. The data were obtained from November 2018 to December 2019. The process of tool design and validation are explained in (Figure 1)

### 2.2. Questionnaire Construction

The Primary Care Assessment Survey and the Primary Care Assessment Tool scales [9,10], in addition to the EAA subscale, were used as a base for the questionnaire. These scales were used as the base for the questionnaire due to the fact that they were the best validated scales measuring accessibility to primary care in rural and urban areas available and, in the literature, the lack of an Arabic language version that focuses on specific barriers relating to cleft lip and palate patients necessitates the construction of this tool.

Seventeen items that related to barriers specific to cleft lip and plate patients were extracted and presented for translation. The items related to various aspects that patients with OFCs, and their families, face during treatment, such as “average waiting time for appointments”, “time needed to get to treatment centers”. For the seventeen items, the questions were structured as a five-point Likert scale ranging from 1 to 5, with “1” referencing having extreme difficulty and “5” referencing having the least amount of difficulty. It was translated by a certified bilingual translator and the questions were modified to fit the needs of cleft lip and palate patients. It was then translated back to English by a certified bilingual translator, compared with the Arabic version, and subjected to validation procedures. These procedures included content validity, factor analysis, and internal consistency. 

### 2.3. Content Validity 

Questionnaire content validity was conducted by a calculation of a content validity index (CVI) for each item (Item-CVI (I-CVI)) as well as for the total scale (Scale-CVI (S-CVI)). A panel of six independent experts reviewed the questionnaire and rated each item based on relevance (R), clarity (C), simplicity (S), and ambiguity (A) using a 4-point Likert type scale from 1 to 4, with 1 signifying “not clear at all” and 4 “very clear” [11], where higher scores indicated a better index. Other ratings, including clarity, simplicity, and ambiguity, were used to modify the formulation or answering options of the item, if it was deemed relevant. S-CVI was calculated as the proportion of items rated as 3 or 4 by all raters, that is, the proportion of items scored as 1 on the relevance scale. An S-CIV ≥ 0.8 was considered for scale relevance.

### 2.4. Data Collection

Data collection was done via phone interview. The phone number of one of the parents of each patient was obtained from the hospital records and they were interviewed. The data were entered immediately in an electronic version of the questionnaire.

### 2.5. Statistical Methods

SAS version 9.4 (SAS Institute Inc., Cary, NC, USA) was used to analyze the data. To assess the reliability and validity of the questionnaire, it was subjected to internal consistency, content validity, and factor analysis.

### 2.6. Factor Analysis

In this study, orthogonal rotation of the factors was performed using varimax rotation. Factor analysis was performed to determine the factor structure of the instrument. In addition, it determined which factor accounted for the majority of variance in healthcare barriers for cleft lip and palate patients. Factor analysis is used to determine how a subset of items is related by using the correlation matrix between items on a scale that suggests that they are measuring the general concept of interest [12]. Principal component analysis extracts factors and keeps in the first factor the maximum possible amount of common variance. Subsequent factors keep the maximum amount of the remaining common variance until all common variance is included [12]. Factors are always listed according to the amount of variation they explain in descending order (i.e., from the highest (first factor) to the lowest (last factor)).

An eigenvalue shows the amount of variance explained by each factor. Eigenvalues above 1.00 are considered strong enough to be kept [12]. Within each factor, item loading was categorized as follows: >0.70 excellent, >0.63 very good, >0.55 good, >0.45 fair, and >0.32 poor [10]. For each item, the highest loading in a factor was considered. To determine sampling adequacy, the Kaiser-Meyer-Olkin measure (KMO) was used. A KMO value equal to 0.70 indicates that factor analysis can be performed. 

### 2.7. Reliability

Cronbach’s alpha was used to test internal consistency. The significance level was set at *p* < 0.05. 

## 3. Results

A total of 165 parents were interviewed, including 100 males and 65 females. Of the parents, 90.3% were Saudi citizens. The sample was divided almost equally between urban and rural environments, amounting to 46.7% and 49.7%, respectively. Our results showed that 76.4% were cleft lip and palate patients, 9.7% were cleft palate only patients, and 13.3% were cleft lip only patients. 

The percentage of patients who traveled for appointments was 61.8% and 57% stated that the treatment center was more than an hour away. The average waiting time for appointments was more than 6 weeks according to 89.7% of the respondents, and 47.3% found scheduling appointments very difficult.

### 3.1. Content Validity 

I-CVI was calculated as the proportion of raters who gave a 3 or 4 relevance score to a given item and the item was deemed relevant for an I-CVI score = or > 0.80. 

Rating of the first version of the questionnaire (17 items) resulted in S-CVI = 0.65, indicating non-acceptance of the scale. Results of I-CVIs as well as the mean item ratings of the first scale version are depicted in Table 2. 

Based on the previous results, including I-CVIs and other ratings, as well as raters’ suggestions, the scale was modified by deleting irrelevant items and reformulating the relevant ones which had clarity, simplicity, and/or ambiguity issues. The updated version of the questionnaire included nine questions divided into three sections: (A) geographic accessibility (three items); (B) difficult accessibility (four items); and (C) appointment accessibility (two items). This updated version of the questionnaire underwent the same content validation process by a panel of five experts—two experts who had been included before, and three new ones. Rating results showed I-CVI = 1 (for all nine items, indicating S-CVI = 1). Thus, this updated version was adopted as the final version of the scale using the same nine-item scale.

### 3.2. Principal Component Factor Analysis 

The KMO value was 0.67 above the minimum acceptable value of 0.6 (Table 3). The factor structure after varimax rotation is shown in Table 3. There were three factors with eigenvalues above 1.00, which collectively accounted for 73% of the variance.

The factors were as follows: Factor 1: geographic accessibility; Factor 2: appointment availability and accessibility; Factor 3: scheduling-related barriers. Items in the first factor included questions from 1–3 that aimed to assess the travel needs, distance, and time required to reach the specialized care center with an item loading of 0.908, 0.917, and 0.897, respectively, with an eigenvalue of 3.083. The second factor included question numbers 6, 9, 10, and 11 with an eigenvalue of 1.935. These questions targeted appointment accessibility and availability barriers with a factor loading of 0.631 for question 6, 0.847 for question 9, 0.845 for question 10, and 0.705 for question 11. The scheduling-related barrier questions were questions 4 and 5 with a factor loading of 0.838 and 0.879, respectively, and an eigenvalue of 1.529 (Table 4). 

### 3.3. Reliability

Cronbach’s alpha was used to test internal consistency. Cronbach’s alpha was 0.86. 

## 4. Discussion

It is important to understand and measure the healthcare needs of patients, especially for a group that requires specialized attention, such as patients with oral clefts. In order for these difficulties to be correctly and accurately measured, a tool must be available that is both validated and comprehensive. 

When we look at this tool, we can see that the index seems to be both valid and reliable according to its high CVI score and Cronbach’s alpha score.

In this study, factor analysis was performed to determine the factor structure of the instrument. An additional reason for doing this was to determine the factor accounting for the majority of variance in healthcare barriers to cleft lip and palate patients. 

Three factors were extracted: Factor 1: geographic accessibility; Factor 2: appointment availability and accessibility; Factor 3: scheduling-related barriers. Items in the first factor (geographic accessibility) were presumed to measure geographic availability and included information regarding the location of the clinic, the simplicity and ease of travel for emergency and regular care, and the time taken to reach the clinic. Items in the second factor (appointment availability and accessibility) measured the difficulty the patients experienced in accessing appointments, e.g., having a general lack of access to appointments or having work or school as a barrier to access. Items in the third factor (scheduling-related barriers) looked at the waiting time for appointments and the time interval between two consecutive scheduled visits to the clinic.

This aspect of validation differentiates this tool and makes it unique with respect to other studies without a validation process or with an evaluation method that is purely qualitative. An example of such studies includes Åstrøm et al., who assessed perceived oral healthcare needs in Tanzanian adults. Although the questionnaire used was assessed by professionals in terms of quality and selection of appropriate vocabulary and cultural appropriateness to the target population, a quantitative assessment was not performed [13]. Hoad-Reddick evaluated perceived dental care needs in the elderly, but they conducted no validation methods [14]. Rungsiyanont et al. evaluated perceived dental needs and attitudes to dental practices in AIDS patients [15].

It is important for society to use such tools to measure barriers to healthcare in order to improve and adjust its healthcare system. This is crucial in a country such as Saudi Arabia, where a single organization, the Ministry of Health (MOH), operates more than 60% of the healthcare services in the country free of charge. This issue is becoming more important in recent times as the population of Saudi Arabia has increased and healthcare costs are escalating exponentially. Awareness around healthcare is growing in the population, causing the demand for specialty health services to rise. The slow construction capacity in Saudi Arabia has led to a huge deficit with long waiting times and difficulty in obtaining optimal care [16]. According to Al Shamsi et al., problems relating to referrals to secondary/specialist care are one of the main barriers to accessing care in Saudi Arabia [17].

A crucial consideration when discussing barriers to care is how they are assessed by each individual, and the psychological impact of these barriers on each individual. The interpretation of a difficulty or an obstacle may vary due to several factors such as age, cognitive level, gender, temperament, and cultural background [18]. Therefore, a myriad of situational and individual issues might not be revealed by means of a questionnaire. This should be taken into consideration when interpreting data gathered by such a tool.

When we look at the literature, we can see a lack of Arab participation in research about barriers to care among patients with cleft lip and palate. By using this instrument in future research, barriers to seeking healthcare among the Arab population can be identified, and Arabic participation in such studies can be enriched.

This study has several limitations. One of the limitations is that the data collection was done via phone interview, which eliminated the participation of patients with no phones or disconnected phone numbers. In our study, about one fifth of the participants did not answer the phone or had disconnected phone numbers. Further studies that use other methods of data gathering such as direct interviews or surveys by mail can cover the subset of the population with no access to phones. In addition, the study is limited to the Kingdom of Saudi Arabia, excluding the other 22 Arabic-speaking countries. Future studies should include subjects from other Arabic countries. A limitation of the scale is its generalizability in other countries. Since the scale was developed in Saudi Arabia, the scale might not capture a number of barriers observed in different cultures. Further studies with larger sample sizes and different Arabic-speaking countries are recommended.

## 5. Conclusions

Three factors related to barriers facing patients with OFCs and their families were identified, indicating that this instrument has a simple factor structure. Based on this study, it can be concluded that the tool is representative of the barriers facing patients with OFCs, and is therefore capable of measuring them. It should be reapplied in healthcare contexts to strengthen the care provided to patients with OFCs in particular, and to the community as a whole.

## Figures and Tables

**Figure 1 ijerph-18-03399-f001:**
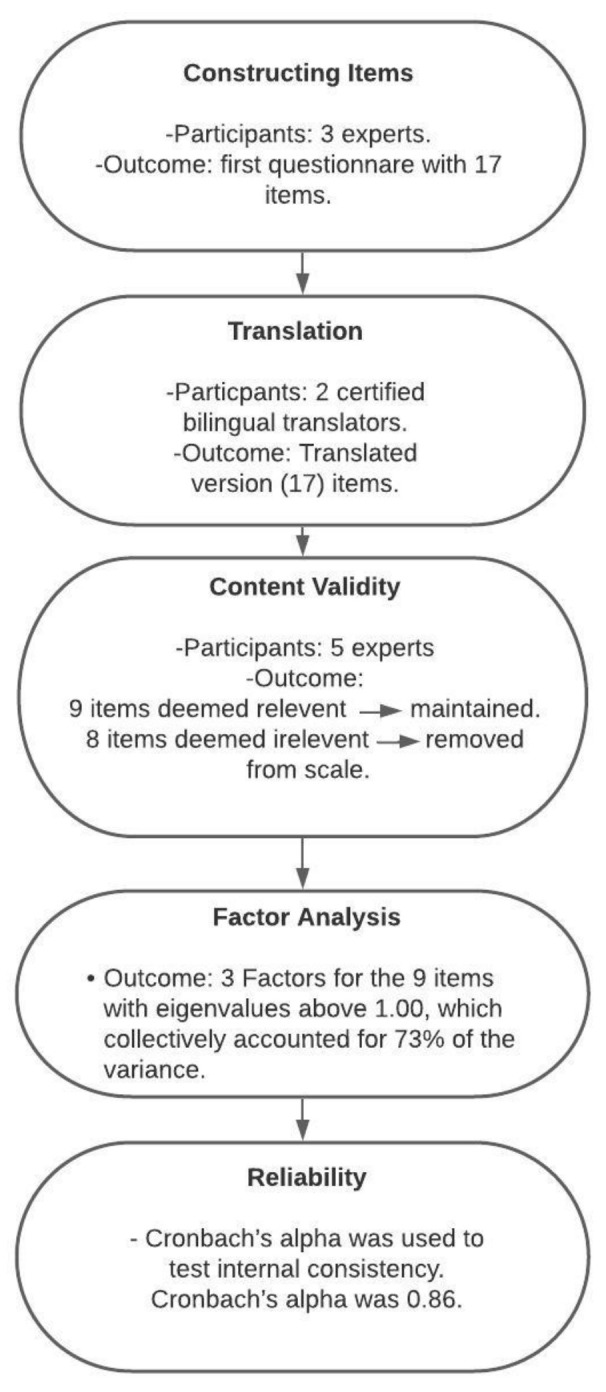
Flowchart of the tool design and validation process.

**Table 1 ijerph-18-03399-t001:** List of tertiary medical centers included in the study according to their city and regional location.

City	Center	North	South	West	East
Jeddah	King Fahad Hospital	King Faisal Specialized Hospital and Research Centre	King Abdulaziz Medical City	King Fahad Armed Hospital	King Abdulaziz University Hospital
Riyadh	King Fahad Medical City	King Faisal Specialized Hospital and Research Centre	Alshemaisy Hospital	Riyadh Military Hospital	Riyadh National Guard Hospital
Madinah	Madinah Maternity and Children’s Hospital				

**Table 2 ijerph-18-03399-t002:** Item rating and content validity index of the first questionnaire version.

Item	Mean Rating				I-CVI			
	R	C	S	A	R	C	S	A
1	4	3.8	4	3.8	1.00	1.00	1.00	1.00
2	3.6	4	4	4	0.80	1.00	1.00	1.00
3	4	4	3.4	4	1.00	1.00	0.80	1.00
4	4	3.6	4	3.8	1.00	1.00	1.00	1.00
5	2.6	3	3.2	3.2	0.40	0.60	0.60	0.60
6	4	3.8	4	4	1.00	1.00	1.00	1.00
7	4	3.8	4	4	1.00	1.00	1.00	1.00
8	3.6	2.8	2.8	3.6	0.80	0.40	0.40	0.80
9	3.6	3.8	3.8	4	0.80	1.00	1.00	1.00
10	3.6	3.4	3.4	3.2	1.00	1.00	1.00	0.80
11	3.6	3.4	3.4	3.4	0.80	0.80	0.80	0.80
12	3.6	3.2	3.2	2.8	1.00	0.80	0.80	0.60
13	3.6	3.6	3.6	3.6	1.00	1.00	1.00	1.00
14	3.6	4	4	3.4	0.80	1.00	1.00	0.80
15	4	3.8	4	4	1.00	1.00	1.00	1.00
16	3.8	3.8	3.8	3.8	1.00	1.00	1.00	1.00
17	4	4	4	4	1.00	1.00	1.00	1.00

I-CVI: Item content validity index; R: relevance; C: clarity; S: simplicity; A: ambiguity; S-CVI: scale content validity index; S-CVI = 0.65.

**Table 3 ijerph-18-03399-t003:** Kaiser’s measure of sampling adequacy.

Kaiser’s Measure of Sampling Adequacy: Overall Measure of Sampling Adequacy (MSA) = 0.66557747								
q1	q2	q3	q4	q5	q6	q9	q10	q11
0.76113703	0.67770447	0.70750641	0.54590162	0.50910000	0.71482085	0.66615379	0.63683160	0.60300651

**Table 4 ijerph-18-03399-t004:** Factor analysis.

Rotated Factor Pattern (3 Factors)				
		Factor 1	Factor 2	Factor 3
q1	First question: Do you travel for your appointments?	0.908 *	0.119	−0.053
q2	Second question: How far is the specialized care center from your residence?	0.917 *	0.072	−0.053
q3	Third question: How much time do you need to get to the specialized clinic?	0.897 *	0.120	0.078
q4	Fourth question: What is the average waiting time for your physician’s appointment?	−0.049	−0.217	0.838 *
q5	Fifth question: How long is the interval between two consecutive consultations?	0.023	0.105	0.879 *
q6	Sixth question: How easy/difficult is it to obtain a school leave for your diseased child’s medical appointments?	0.069	0.631 *	0.115
q9	Ninth question: How many appointments have you missed because of your work?	0.154	0.847 *	−0.104
q10	Tenth question: How many appointments have you missed because of your child’s school?	0.029	0.845 *	−0.237
q11	Eleventh question: How many appointments have you missed because you could not find caregiver for your other children?	0.113	0.705 *	0.131
	Eigenvalue	3.083	1.935	1.529
	% of explained variance	34.25	21.50	16.99

* The highest loading for each item is presented in boldface. Factor 1: geographic accessibility. Factor 2: appointment availability and accessibility. Factor 3: scheduling-related barriers.

## Data Availability

Data is available upon request form the Author.

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
