# Peer review of "Instrument Development and Validation to Assess Care Barriers for Patients in Saudi Arabia with Oral Clefts"

_ijerph, 2021, doi:10.3390/ijerph18073399_

Round 1

Reviewer 1 Report

I read the manuscript ijerph-1130713 and here are my comments:

Materials and Methods: more clear description of construction and content validity process is required.

Questionnaire Construction: The authors have chosen two existing instruments; The Primary Care Assessment Survey and the Primary Care Assessment Tool, to develop new instrument to assess care barriers for patients in Saudi Arabia with oral clefts. At the same time there is a statement in the introduction that these scales cannot be used for Arabic speaking patients with cleft lip and palate conditions. It is therefore a need for more detailed explanation about these instruments relevance and the choice of these instruments as the base for the questionnaire.

Furthermore, it is stated in the materials and methods that seventeen items were extracted and presented for translation. Better description about total number of items and why the seventeen items were chosen should be given by the authors.

Content Validity: It is described in the manuscript that the questions were structured as a five point Likert scale ranging from 1 to 5. It is not described the definition of 1,2, 3, 4 and 5 in the Likert scale chosen for the study. It is also difficult to understand if the Likert scale 1-5 is for the seventeen items or for the nine final questions.

Results:

In the Material and Methods section, the authors refers to seventeen and elven items. However, in the results it is stated that the updated version of the questionnaire included nine questions divided into three sections (Page 5, Line 158). It is difficult to follow if the items are the same as the previous described items or if the sections are the items.

Page 5, Line 160-162: In this sentence there is description  content validation process of the updated version of the questionnaire. The authors describe that this process is done for all eight items. However, in the sentence before the authors describe 9 items being included in the updated version of the questionnaire. The number of items is either incorrect or the description of the process not correctly described.

Other comments:

The introduction and discussion sections are rather short and there are only 13 references.

Page 2, Line 51-52: The sentence is difficult to understand. Can be shortened.

Page 4, Line 134: “A total of 165 parents were interviews” should be change to “A total of 165 parents were interviewed

Page 3, Line 90-91: It should be added to the sentence that description on how content validity, factor analysis, and internal consistency procedures had been conducted follows below.

Author Response

Thank you for your comments dated 10/3/2021. We have carefully reviewed the comments and have revised the manuscript accordingly. Our responses are given in a point-by-point manner. Changes to the manuscript are highlighted in yellow. 

We hope the revised version is now suitable for publication and look forward to hearing from you in due course. 

Sincerely, 

Layla Khogeer

Pediatric dentistry department,  King Abdulaziz Universty.

Reviewer 2 Report

This paper aims to understand the instrument development and validation to assess care barriers for patients in Saudi Arabia with oral clefts. I think that the data from this study are valuable. Therefore, I believe that this study will be of interest to the journal’s readership. However, because of some major concerns and shortcomings in the content, I wish to call your attention to the following:

1) The authors obtained the data via phoning. I think this method can affect the results in this study. The authors mentioned some shortcoming in limitation, but the influence should be discussed. For example, how many subjects did not have telephone number? If there were lots of people without telephone number, the participants may be specific population.

2) There authors should explain the characteristics of the participants and the patients.

Have the oral clefts treatment the patients received finished? Please give me the information about the treatment phase. If you target the participants in different treatment phase, I recommend you discuss that.

3) Please confirm the period of data collection. Was the second study conducted before the first study?

4) The tables did not have appropriate foot notes. Please revised the tables.

5) In the second paragraph in Discussion section, what does the “(1)” mean?

Author Response

Thank you for your comments dated 10/3/2021. We have carefully reviewed the comments and have revised the manuscript accordingly. Our responses are given in a point-by-point manner. Changes to the manuscript are highlighted in yellow. 

We hope the revised version is now suitable for publication and look forward to hearing from you in due course. 

Sincerely, 

Layla Khogeer

Pediatric dentistry department,  King Abdulaziz Universty

Round 2

Reviewer 2 Report

Thank you for revising the manuscript.

The authors dealt with my comments well.

I think that this manuscript is suitable for publication. 

Thank you for the opportunity to read this valuable manuscript.